# Multimodal Knowledge Distillation for Emotion Recognition

**DOI:** 10.3390/brainsci15070707

**Published:** 2025-06-30

**Authors:** Zhenxuan Zhang, Guanyu Lu

**Affiliations:** 1State Key Laboratory of Media Convergence and Communication, Communication University of China, Beijing 100024, China; 2022200810011071@cuc.edu.cn; 2School of Cyberspace Security, Beijing Jiaotong University, Beijing 100044, China

**Keywords:** multimodal knowledge distillation, physiological signals, emotion recognition, deep learning, emotion BCI

## Abstract

Multimodal emotion recognition has emerged as a prominent field in affective computing, offering superior performance compared to single-modality methods. Among various physiological signals, EEG signals and EOG data are highly valued for their complementary strengths in emotion recognition. However, the practical application of EEG-based approaches is often hindered by high costs and operational complexity, making EOG a more feasible alternative in real-world scenarios. To address this limitation, this study introduces a novel framework for multimodal knowledge distillation, designed to improve the practicality of emotion decoding while maintaining high accuracy, with the framework including a multimodal fusion module to extract and integrate interactive and heterogeneous features, and a unimodal student model structurally aligned with the multimodal teacher model for better knowledge alignment. The framework combines EEG and EOG signals into a unified model and distills the fused multimodal features into a simplified EOG-only model. To facilitate efficient knowledge transfer, the approach incorporates a dynamic feedback mechanism that adjusts the guidance provided by the multimodal model to the unimodal model during the distillation process based on performance metrics. The proposed method was comprehensively evaluated on two datasets based on EEG and EOG signals. The accuracy of the valence and arousal of the proposed model in the DEAP dataset are 70.38% and 60.41%, respectively. The accuracy of valence and arousal in the BJTU-Emotion dataset are 61.31% and 60.31%, respectively. The proposed method achieves state-of-the-art classification performance compared to the baseline method, with statistically significant improvements confirmed by paired *t*-tests (*p* < 0.05), and the framework effectively transfers knowledge from multimodal models to unimodal EOG models, enhancing the practicality of emotion recognition while maintaining high accuracy, thus expanding the applicability of emotion recognition in real-world scenarios.

## 1. Introduction

Emotion computing refers to the use of computational techniques to identify, analyze, interpret, and express human emotions. It is an interdisciplinary field encompassing artificial intelligence, computer science, psychology, and neuroscience. Emotion recognition, as a critical branch of emotion computing, focuses on identifying and classifying human emotional states through various signals. It plays an essential role in enhancing human–computer interaction and expanding the application scenarios of intelligent systems.

Emotion recognition methods can be broadly divided into two categories: physiological-signal-based methods and behavior-based methods. Physiological-signal-based methods use physiological data, such as electroencephalogram (EEG) and electrooculogram (EOG), as input features for emotion recognition. These signals are more stable than external behavioral features like facial expressions or speech. Consequently, physiological signals are increasingly used in emotion recognition systems due to their reliability in complex scenarios.

Human emotions involve intricate physiological processes, making single-modal signals insufficient for effective emotion recognition. As a result, researchers have increasingly investigated the use of multimodal data for emotion recognition [1,2]. These studies predominantly explore the interplay and diversity among various modalities. Regarding heterogeneity, the focus is on analyzing unique signal features within individual modalities. For interactivity, the emphasis lies on understanding the complementary aspects of physiological signals across different modalities. Findings from such studies indicate that multimodal data offer a more holistic representation of human emotions, enhancing recognition accuracy [3]. Consequently, multimodal emotion recognition has emerged as a pivotal area of interest in recent research.

Although multimodal physiological signals can improve the performance of models, the challenge lies in the complexity of the data collection process during experiments. For example, participants watching movie clips might be distracted, making it difficult for them to stay relaxed during testing, thereby affecting their psychological responses. Additionally, the collection of EEG signals imposes high environmental requirements, requiring a stable environment for data acquisition, which limits the application of emotion recognition in real-world scenarios. However, compared to the more challenging EEG signals, EOG signals can be collected with just one or a few simple sensors, making the data acquisition process much simpler. Moreover, EOG signals are less constrained by participants’ activities during the collection process, providing more authentic signals while being more cost-effective and efficient for experiments. However, relying solely on unimodal EOG signals for model construction is less effective than using multimodal data models. Therefore, how to leverage knowledge distillation techniques to transfer the rich knowledge from multimodal features (teacher model) to unimodal models (student model) becomes a critical issue. This approach helps enhance the generalization and practicality of unimodal models.

This paper proposes a novel emotion recognition framework based on physiological signals, which integrates a multimodal fusion module and a dynamic feedback regulation mechanism. The framework enables the unimodal student model to effectively learn interactive and heterogeneous knowledge from a multimodal teacher model. Specifically, the proposed method leverages EEG and EOG signals for emotion recognition. The primary contributions of this paper are as follows:A multimodal fusion module is designed to extract and integrate interactive and heterogeneous features, enabling the unimodal student model to learn from multimodal distributions.A dynamic feedback regulation mechanism is introduced, which continuously adjusts the teacher model based on the performance of the student model, thereby enhancing the effectiveness of knowledge transfer.Experimental results on the emotion datasets demonstrate that the proposed method achieves superior performance compared to baseline models, expanding the applicability of emotion recognition.

The remainder of this paper is structured as follows: Section 2 introduces related work, including multimodal emotion recognition and knowledge distillation. Section 3 describes the proposed method, detailing the multimodal fusion module and the dynamic feedback regulation mechanism. Section 4 presents experimental results and analysis, while Section 5 concludes the paper and discusses future research directions.

## 2. Related Work

### 2.1. Multimodal Physiological Signals

Multimodal physiological signals are a combination of various signals derived from different human physiological systems, including but not limited to electroencephalograms (EEG), electrooculograms (EOG), electromyograms (EMG), and others [4]. These signals encapsulate a wide spectrum of physiological states, such as emotions, cognitive conditions, and mental illnesses, making them invaluable in fields like medicine, psychology, and neuroscience. Multimodal physiological signals can be acquired through two primary methods: sensor-based acquisition and image processing techniques. For sensor-based acquisition, sensors are placed on the subject to collect physiological signals through electrodes or other devices. In contrast, image processing techniques extract physiological information such as facial expressions and eye movements by analyzing human body images.

The analysis and processing of multimodal physiological signals can be achieved using methods such as digital signal processing and machine learning [5]. For instance, signal processing techniques can be employed to preprocess data and extract features relevant to specific physiological states, which are then classified or predicted using machine learning algorithms. This enables the recognition and prediction of physiological states. Multimodal physiological signals are widely applied in medical diagnostics, neuroscience research, psychology, and human–computer interaction. One prominent application is emotion recognition, where different emotional states—such as joy, sadness, and anger—can be identified, supporting mental health assessment, emotional intervention, and interactive systems.

This study primarily uses EEG signals and EOG signals for emotion recognition. EEG signals refer to the bioelectrical signals of brain activity. These signals are obtained by placing electrodes on the scalp to measure and record the electrical activity of neurons in the cerebral cortex. EEG signals are widely used to study various physiological and pathological brain states, such as sleep, attention, epilepsy, and brain injuries. The frequency range of EEG signals generally lies between 0.5 Hz and 100 Hz, with the most prominent frequencies being in the 4 Hz to 30 Hz range, including alpha, beta, and gamma waves.

EOG signals, on the other hand, are bioelectrical signals that measure eye movements. These signals are generated by the difference in potential between the cornea and retina, allowing the tracking of horizontal and vertical eye movements. EOG is particularly useful for detecting visual attention, cognitive load, and emotional states, as eye movement patterns often reflect changes in these aspects. Compared to EEG signals, EOG signals are easier to acquire and require fewer electrodes, making them less intrusive and more practical for real-world applications.

By combining EEG and EOG signals, this study can achieve a more comprehensive understanding of participants’ emotional and psychological states, thereby enhancing the accuracy and applicability of emotion recognition systems.

Interactivity and heterogeneity are two key concepts in the analysis of multimodal physiological signals. Interactivity refers to the relationships and dependencies between different physiological signals. For example, EEG and ECG signals may interact due to the complex neurofeedback mechanisms between the brain and the heart. Such interactions require in-depth analysis and modeling to capture the dynamics of physiological systems. Heterogeneity, on the other hand, refers to the distinct characteristics of different physiological signals. For instance, EEG and EMG signals exhibit different frequency and amplitude features, necessitating tailored signal processing and modeling techniques. This heterogeneity introduces additional complexity to the analysis of multimodal physiological signals.

In multimodal physiological signal analysis, it is essential to account for both interactivity and heterogeneity. Modeling the interactivity between signals can reveal the intricate dynamics of physiological systems and enhance the overall analysis. Simultaneously, considering heterogeneity allows for the utilization of unique signal features, enriching the information and expressive power of the multimodal approach. In this study, we develop a neural network to model both the interactivity and heterogeneity of EEG and EOG signals, achieving superior performance. Additionally, we use knowledge distillation to transfer these properties, thereby enhancing the generalization ability of the unimodal model.

In conclusion, multimodal physiological signals hold immense potential due to their rich information content and wide applicability. With ongoing technological advancements and expanding use cases, the analysis and processing of these signals will continue to find broader and deeper applications across diverse fields.

### 2.2. Emotion Recognition Utilizing Physiological Signals

In recent years, emotion recognition has garnered increasing attention in the fields of brain–computer interfaces (BCIs) and intelligent healthcare. To address the challenges posed by individual variability and the complexity of multimodal signals, a range of advanced models have been proposed. Some studies exploit the intrinsic neighborhood semantic structure of EEG signals to facilitate domain adaptation for cross-subject generalization [6]. Others adopt dynamic heterogeneous graph recurrent neural networks to capture temporal dependencies and intermodal dynamics [7]. Techniques such as self-distillation and dynamic interaction have been introduced to enhance modality-specific feature learning and fusion [8]. Furthermore, hierarchical spatial–temporal learning frameworks based on transformers have been developed to model emotion-related patterns from local brain regions to global activity [9]. In addition, disentangling strategies and meta-learning mechanisms are employed to improve the robustness and adaptability of multimodal emotion recognition systems [10,11]. Collectively, these approaches reflect the current trend of integrating domain adaptation, graph learning, and attention mechanisms to tackle the challenges in cross-subject and multimodal emotion recognition [12].

In recent years, emotion recognition based on physiological signals has gained significant attention from researchers [13]. EEG signals, which directly reflect human brain activity, are widely used in emotional brain–computer interfaces. However, human emotions are complex physiological processes, and researchers have found that multimodal physiological signals, such as the combination of EEG with EOG, and ECG, can enhance emotion recognition performance [14]. Studies have shown that multimodal data fusion achieves better classification results compared to unimodal approaches [15,16,17,18]. This has led to extensive research on modeling the heterogeneity and interaction between multimodal signals [19]. For instance, Liu et al. [20,21] utilized a multimodal self-encoder and attention-based strategies to model interactions, while Guo et al. [22] and Song et al. [23] focused on modeling heterogeneity using CNN-based architectures and voting mechanisms.

Despite these advancements, most methods fail to simultaneously address the interaction and heterogeneity of multimodal data and predominantly rely on EEG signals combined with peripheral signals. This reliance on EEG presents practical challenges, as high-quality EEG data acquisition is cumbersome and costly [24,25]. EEG setups often require subjects to wear restrictive caps and maintain fixed positions in controlled environments, making real-world applications challenging. In contrast, EOG signals offer advantages such as easier acquisition and lower equipment costs while maintaining a strong correlation with human emotions [26,27]. Researchers have explored various methods to leverage EOG for emotion recognition, including wavelet-based denoising and SVM classification [28], quantum neural networks optimized with particle swarm algorithms [29], and deep hybrid neural networks [30].

Overall, sentiment recognition results using unimodal modalities are usually lower than those using multimodal modalities, due to the fact that multimodal data usually contain more latent knowledge. Therefore, it is important to know how to effectively transfer the knowledge captured in a large multimodal model to a lightweight unimodal model. The multimodal knowledge distillation proposed in this paper effectively extracts the intermodal heterogeneity and interaction and transfers them to the unimodal model through knowledge transfer. On the one hand, the multimodal information can be effectively utilized to better enhance the performance of the unimodal model, and on the other hand, it can alleviate the many limitations of multimodality in practical applications, and thus promote the grounded application of emotion recognition [31].

## 3. Proposed Methodology

As shown in Figure 1, the multimodal knowledge distillation framework proposed in this paper contains three main components: the multimodal teacher model, the unimodal student model, and the dynamically regulated knowledge-based distillation. The multimodal teacher model captures the knowledge of multimodal physiological signals and incorporates multimodal features to achieve better classification performance. In this paper, we propose a dynamic feedback-regulated knowledge distillation approach that transfers the knowledge captured by the multimodal teacher model to the unimodal student model.

### 3.1. Multimodal Teacher Model

The multimodal teacher model starts by using two convolutional layers to process the raw physiological signals. For an input signal *X* with ch channels and a sampling rate of *f*, the convolutional filters are expressed as(1)$$x_{\mathrm{Filter}\;\mathrm{output}}=CNN_{\left(ch_{1},f_{1}\right)}\left(CNN_{\left(ch_{2},f_{2}\right)}\left(X\right)\right)$$

In this equation, $CNN_{\left(x,y\right)}$ refers to a convolutional layer with a kernel size of (x,y). The resulting feature maps are then passed to a transformer encoder, which extracts the heterogeneous features from different modalities and uses fusion modules to strengthen interactions.

#### 3.1.1. Transformer-Based Feature Extraction for Modal Heterogeneity

To effectively capture the heterogeneous characteristics across multiple modalities, this research utilizes a multi-layer transformer encoder to extract features, mathematically described as(2)$$\mathrm{MultiHead}\left(Q,K,V\right)=\mathrm{Concat}\left({\mathrm{head}}_{1},\ldots ,{\mathrm{head}}_{h}\right)W^{O}$$
where *h* indicates the number of attention heads, *Q* represents the query matrix, *K* is the key matrix, and *V* is the value matrix. Matrix $W^{O}$ provides the output weights. Each individual attention head ${\mathrm{head}}_{m}$ is calculated as(3)$${\mathrm{head}}_{m}=\mathrm{Attention}\left(QW_{m}^{Q},KW_{m}^{K},VW_{m}^{V}\right)$$
where $W_{m}^{Q}$, $W_{m}^{K}$, and $W_{m}^{V}$ are the respective weight matrices for *Q*, *K*, and *V*. The attention mechanism Attention(q,k,v) is further defined as(4)$${\mathrm{head}}_{m}=\mathrm{Attention}\left(QW_{m}^{Q},KW_{m}^{K},VW_{m}^{V}\right)$$
where $W_{m}^{Q}$, $W_{m}^{K}$, and $W_{m}^{V}$ are the weight matrices for *Q*, *K*, and *V*, respectively. Furthermore, the attention mechanism Attention(q,k,v) is defined as(5)$$\mathrm{Attention}\left(q,k,v\right)=\mathrm{softmax}\left(\frac{qk^{⊤}}{\sqrt{d_{k}}}\right)v$$
where $d_{k}$ indicates the size of the *k*-vector. The encoder’s output vectors are utilized to extract cross-modal interactive features, which are subsequently processed by the downstream transformer encoder for enhanced high-dimensional feature representation.

By implementing a dual-stream, multi-layer transformer encoder architecture, the model achieves more effective extraction of heterogeneous attributes across multiple modalities. This design uncovers hidden high-dimensional features from multimodal fusion, ultimately supporting unimodal models in delivering enhanced performance.

#### 3.1.2. Modal Fusion Module for Interactivity Features

To effectively capture cross-modal interactions and organize multimodal features systematically, this study introduces a fusion module leveraging interaction fractions. Initially, high-dimensional features from various modalities are combined through concatenation to form unified representations. Interaction extractors are then employed to compute interaction fractions, with the outputs weighted and aggregated to produce the final fused features.

The four heterogeneous features in this module, denoted as $X_{1}^{\mathrm{EOG}},X_{2}^{\mathrm{EOG}},X_{1}^{\mathrm{EEG}},$ and $X_{2}^{\mathrm{EEG}}$, correspond to the four inputs of the fusion module. The operation of this fusion module is formally defined as(6)$$X_{\mathrm{output}}^{\mathrm{IMF}}=IE_{1}\left(\mathrm{Concat}\left(X_{1}^{\mathrm{EOG}},X_{1}^{\mathrm{EEG}}\right)\right)⊕IE_{2}\left(\mathrm{Concat}\left(X_{2}^{\mathrm{EOG}},X_{2}^{\mathrm{EEG}}\right)\right)$$
where $IE_{l}$ represents the *l*-th interaction extractor, Concat denotes the concatenation operation for tensors, and ⊕ represents the element-wise addition of two feature tensors. For $X_{i}^{M}$, subscript *i* corresponds to one of the four inputs, and *M* represents the modality.

In this paper, an interaction extractor is utilized, where the input primarily consists of concatenated multimodal features. The computation in the interaction extraction module is defined as(7)$${\mathrm{Score}}_{\mathrm{Interactivity}}=\sigma \left({\mathrm{FC}}_{2}\left(\delta \left({\mathrm{FC}}_{1}\left(X_{\mathrm{input}}^{IE}\right)\right)\right)\right)$$
where the interaction score is multiplied with the input features element-wise to generate the final output $X_{\mathrm{output}}^{IE}$.(8)$$X_{\mathrm{output}}^{IE}={\mathrm{Score}}_{\mathrm{Interactivity}}⊙X_{\mathrm{input}}^{IE}$$

In summary, by effectively extracting heterogeneous and interactive features, the proposed model dynamically integrates multimodal fusion during training. This allows the extraction of high-quality features and improves the performance of downstream classification tasks, while also providing better guidance to unimodal models and enhancing their performance.

### 3.2. Unimodal Student Model

In this work, the process of distilling a unimodal student model introduces a fusion module based on heterogeneous feature extraction and weighting. This approach aims to reduce the structural gap between the unimodal student model and the multimodal teacher model.

The structure of the heterogeneous feature extractor is consistent with the interaction extractor introduced earlier. In this work, three features extracted from different positions of the transformer encoder output, denoted as $X_{k}$, where k∈{1,2,3}, are input into the HDF (Heterogeneous Deep Fusion) module. The HDF module processes the inputs in a manner similar to the fusion module described previously. The calculation for the output of the HDF module is as follows:(9)$$X_{\mathrm{output}}^{\mathrm{HDF}}=HE_{1}\left(X_{1}\right)⊕HE_{2}\left(X_{2}\right)⊕HE_{3}\left(X_{3}\right)$$
where $HE_{m}$ represents the *m*-th heterogeneous extractor. The heterogeneous score, ${\mathrm{Score}}_{\mathrm{Heterogeneity}}$, is computed using the same operations described in the interaction extractor, as follows:(10)$${\mathrm{Score}}_{\mathrm{Heterogeneity}}=\sigma \left({\mathrm{FC}}_{2}\left(\delta \left({\mathrm{FC}}_{1}\left(X_{\mathrm{input}}^{IE}\right)\right)\right)\right)$$

Subsequently, the output of the heterogeneous extractor, $X_{\mathrm{output}}^{HE}$, can be expressed as(11)$$X_{\mathrm{output}}^{HE}={\mathrm{Score}}_{\mathrm{Heterogeneity}}⊙X_{\mathrm{input}}^{HE}$$

In summary, this work designed a unimodal student model that is structurally more similar to the multimodal teacher model, facilitating better alignment with the outputs of the teacher model.

### 3.3. Dynamic Feedback Mechanisms

Traditional knowledge distillation methods often rely on a static teacher model, which limits adaptability during knowledge transfer [32]. To overcome this limitation, a dynamic feedback regulation mechanism is introduced, allowing the teacher model to adaptively adjust based on the student model’s performance. This enhances the overall efficiency of knowledge transfer. By extending the conventional student–teacher training framework, a feedback adjustment phase is incorporated to better align the teacher model with the student model, thereby improving knowledge transfer and facilitating the student’s learning process.

In this study, the teacher model is iteratively adjusted throughout the training process to produce knowledge that aligns more effectively with the student model, boosting its performance. The proposed training approach is divided into two main stages: the teaching phase and the feedback adjustment phase. During the teaching phase, the student model utilizes one-hot encoded labels from the pretrained teacher model for guidance. After a set number of iterations, the process shifts to the feedback adjustment phase, where the teacher model is updated based on feedback from the student. This iterative cycle alternates between the two phases until the unimodal student model achieves satisfactory performance. Finally, during the testing phase, the trained student model is evaluated to produce the final outcomes. Subsequent sections delve into the specifics of each training stage.

#### 3.3.1. Teaching Phase

In this phase, the parameters $\theta_{s}$ of the student model are exclusively updated. The objective function is formulated using the cross-entropy loss $L_{\mathrm{CE}}$, which serves to optimize classification accuracy. Given a one-hot encoded label $y=[y_{1},y_{2},\ldots ,y_{K}]$, where *K* represents the total number of classes, and the predicted probabilities $\hat{y}=\left[{\hat{y}}_{1},{\hat{y}}_{2},\ldots ,{\hat{y}}_{K}\right]$, the cross-entropy loss $L_{\mathrm{CE}}$ is expressed as(12)$$L_{CE}\left(y,\hat{y}\right)=-\sum_{k=1}^{K}y_{k}\log\left({\hat{y}}_{k}\right)$$

Simultaneously, the unimodal student model interacts with the teacher model to learn heterogeneous features and enhance classification performance. To ensure this, the output features of the student model, $F_{s}=\left\{f_{1},f_{2},\ldots ,f_{N}\right\}$, must align as closely as possible with the heterogeneous features of the teacher model, $F_{T}=\left\{f_{1}^{T},f_{2}^{T},\ldots ,f_{N}^{T}\right\}$, where *N* represents the number of data points for each feature. To achieve this alignment, a mean squared error (*MSE*) loss is incorporated into the objective function [33]:(13)$$L_{MSE}=\frac{1}{N}\sum_{n=1}^{N}{\left(f_{n}-f_{n}^{\prime }\right)}^{2}$$

Additionally, the output of the student model must align with that of the teacher model. To achieve this, Kullback–Leibler (*KL*) divergence is incorporated as an auxiliary loss term to optimize the student model’s learning process. In assuming that the classification output of the teacher model is $y^{\prime }=\left[y_{1}^{\prime },y_{2}^{\prime },\ldots ,y_{K}^{\prime }\right]$, the *KL* divergence loss $L_{\mathrm{KL}}$ is defined as(14)$$L_{KL}=-\sum_{k=1}^{K}y_{k}\log\left(\frac{y_{k}}{y_{k}^{\prime }}\right)$$

The overall objective function $L_{S}$ combines three weighted loss terms:(15)$$L_{S}=\lambda_{1}L_{\mathrm{CE}}\left(y,f_{S}\left(x;\theta_{s}\right)\right)+\lambda_{2}L_{\mathrm{KL}}\left(f_{T}\left(x;\theta_{T}\right),f_{S}\left(x;\theta_{s}\right)\right)+\lambda_{3}L_{\mathrm{MSE}}\left(F_{S},F_{T}\right)$$
where $\lambda_{1}$, $\lambda_{2}$, and $\lambda_{3}$ are coefficients that balance the contributions of each loss term. Here, $f_{T}\left(x;\theta_{T}\right)$ and $f_{S}\left(x;\theta_{s}\right)$ represent the outputs of the teacher model and the student model, respectively, while *x* denotes the input data. Finally, the optimized parameters of the student model, $\theta_{s}^{*}$, are determined by(16)$$\theta_{s}^{*}=\arg\min_{\theta_{s}}L_{S}$$

#### 3.3.2. Feedback Phase

In the feedback adjustment phase, the parameters $\theta_{s}$ of the student model remain fixed. A temporary student model $f_{s}\left(\cdot ;\theta_{ts}\right)$ is introduced as an auxiliary module, with its parameters copied from the original student model to ensure structural consistency [34]. The primary goal of this phase is to enable the teacher model to adapt its features dynamically according to feedback from the temporary student model, aligning more closely with the performance and characteristics of the student model.

The temporary student model does not directly affect the final student model, but it is vital for optimizing the teacher model. Specifically, it first learns from the teacher model with the help of the loss function $L_{S}$. The updated parameters $\theta_{ts}$ are computed as(17)$$\theta_{ts}=\theta_{s}-\alpha \frac{\partial L_{S}}{\partial \theta_{s}}$$
where α represents the learning rate. Subsequently, using the one-hot encoded labels, the teacher model output, and the IMF features of the updated temporary student model, the teacher model is adjusted to better guide the student model by refining multimodal feature representation. The definitions are as follows:(18)$$\min_{\theta_{t}}L_{CE}\left(y,f_{s}\left(x_{V};\theta_{ts}\right)\right)+\lambda_{t}L_{CE}\left(y,f_{t}\left(x_{V};\theta_{t}\right)\right)$$
where $\lambda_{t}$ represents a scaling factor. In practice, the updated temporary student model and teacher model utilize different training samples. To prevent data leakage, the updated teacher model leverages validation data instead of training data. Since the parameters of the original student model remain unchanged during this phase, there is no risk of data contamination throughout the training process.

In summary, the dynamic feedback adjustment method enables the continuous refinement of the teacher model during knowledge distillation. By selecting appropriate adjustment frequencies and iteration counts, the teacher model dynamically aligns with the evolving optimization needs of the student model [35]. This process effectively guides the student model to better learn multimodal features, ultimately improving its generalization ability and performance on unseen datasets.

## 4. Experimental Results and Analysis

### 4.1. Dataset Description

To comprehensively evaluate the effectiveness of the proposed model, this study uses the emotion datasets. These datasets are widely recognized in the field and provide a large amount of multimodal experimental data with annotated labels. Both datasets play a significant role in emotion computing and analysis.

The DEAP dataset provides various physiological signals and emotional evaluation data, serving as fundamental resources for research in affective computing and physiological signal analysis. Similarly, the BJTU-Emotion dataset also offers multimodal affective data, providing basic resources for affective analysis and data analysis studies. The public availability and standardization of these datasets facilitate advancements in affective computing and multimodal data analysis in related fields.

The DEAP dataset [36] is a multimodal dataset containing EEG, facial expressions, and peripheral physiological signals such as electrooculograms (EOG). It includes data collected from 32 participants while they watched music videos. Their EEG signals, facial expressions, and EOG signals were recorded during these sessions. Each participant completed 40 experiments, with each trial lasting one minute and including a 3 s baseline recording at the start. After each trial, participants provided self-reported ratings on arousal, valence, dominance, and liking on a nine-point scale [37].

BJTU-Emotion is a multimodal dataset we collected. It includes data from 30 participants who were recorded while watching movie clips. The recorded data comprise facial expressions, audio signals, eye gaze data, EEG signals, EOG signals, and other physiological signals. EEG signals were acquired from 32 electrodes following the international 10–20 system, with a sampling rate of 256 Hz. In each trial, participants provided self-reported annotations using integers from 1 to 9 for four dimensions: valence, arousal, dominance, and emotion keywords.

### 4.2. Emotional Dimensions and Evaluation Metrics

#### 4.2.1. Arousal

In emotion recognition, arousal refers to the degree of physiological and psychological activation or stimulation in response to specific stimuli or emotional situations. It measures an individual’s perceived level of energy or excitement, which can range from low to high. A person’s level of arousal can influence their emotions, behavior, and cognition. For example, high arousal is often associated with feelings of anxiety, fear, or excitement, while low arousal is linked to feelings of relaxation or boredom. In emotion recognition tasks, arousal is typically evaluated alongside other dimensions, as it represents the activation level of emotional states, whether pleasant or unpleasant.

#### 4.2.2. Valence

In emotion recognition, valence refers to the subjective experience of the positivity or negativity of an emotional state. It measures the qualitative aspect of an emotion, ranging from extremely positive to extremely negative. For instance, emotions such as happiness, joy, and contentment typically correspond to high valence, whereas emotions such as anger, sadness, and fear are associated with low valence. Valence is commonly evaluated in conjunction with other dimensions, such as arousal, to better understand an individual’s emotional state. Valence is crucial in applications related to mental health and well-being, as it provides insights into the balance of positive and negative emotional experiences over time.

#### 4.2.3. Accuracy

Accuracy is a common evaluation metric in classification tasks. It measures the proportion of correctly classified samples to the total number of samples. For binary classification problems, accuracy is defined as(19)$$\mathrm{Accuracy}=\frac{\mathrm{TP}+\mathrm{TN}}{\mathrm{TP}+\mathrm{TN}+\mathrm{FP}+\mathrm{FN}}$$
where TP represents true positives, TN represents true negatives, FP represents false positives, and FN represents false negatives.

#### 4.2.4. F1 Score

The F1 score balances precision and recall to measure the accuracy of a model on a balanced dataset. When dealing with imbalanced labels, where the number of samples for each label is not equal, the F1 score becomes an important evaluation metric. It combines precision and recall to evaluate the classifier’s performance on imbalanced data. The F1 score is defined as(20)$$\mathrm{F1-score}=\frac{2*\mathrm{TP}}{2*\mathrm{TP}+\mathrm{FP}+\mathrm{FN}}$$
where TP, TN, FP, and FN are as defined above. The F1 score is also used in this paper as a performance metric for classification models.

### 4.3. Experimental Setup

For a fair comparison, the proposed method and baseline methods were evaluated under the same experimental setup. Specifically, since human emotions are a continuous process, the different phases within a single trial are highly correlated. If the dataset is split such that segments within the same trial appear in both the training and validation sets, the model may overfit to trial-specific patterns, causing the test performance to be artificially inflated. However, in real-world applications, a model often needs to predict data from unseen trials that differ significantly from the training data. To mimic this real-world scenario, this paper divides the dataset by trials, ensuring that the training, validation, and test sets are split at an 8:1:1 ratio. After this split, each trial is further segmented into 4 s slices (segments), with each segment treated as an independent sample for model training.

All physiological signals were downsampled to 128 Hz for consistency. Additionally, to prevent overfitting, the final test results were evaluated using the best model checkpoint from the validation phase. All models were implemented using TensorFlow. The HFE’s two fully connected layers had 512 and 256 units, with ReLU and sigmoid activations, respectively, and the IE’s two fully connected layers had 1024 and 512 units, with ReLU and sigmoid activations.

The models were optimized using the Adam optimizer. To improve generalization, all convolutional layers employed dropout regularization during training. Furthermore, in the knowledge distillation phase, the loss function weights were set to $\lambda_{1}$:$\lambda_{2}$:$\lambda_{3}$ = 1:1:1, with the tuning factor $\lambda_{t}=1$ during the feedback phase.

### 4.4. Experimental Results

This paper compares the unimodal student model utilizing EOG features with baseline models. As illustrated in Table 1 and Table 2, the results confirm the effectiveness of the proposed knowledge distillation approach in enhancing the performance of unimodal models. Specifically, the proposed method extracts both interactive and heterogeneous features from the multimodal teacher model through knowledge distillation. This enables the unimodal student model to learn multimodal distributions encompassing both interactive and heterogeneous attributes, leading to significant performance improvements.

In contrast, methods such as DeepConvNet rely exclusively on single-modality signals and fail to utilize multimodal features, thereby limiting their overall performance. Similarly, while CGAN employs knowledge transfer techniques, it overlooks the extraction and integration of heterogeneous features for each modality, further constraining its capabilities. Compared to these approaches, the proposed unimodal student model benefits from joint learning with the multimodal teacher model on both datasets, leveraging the interactive and heterogeneous knowledge transferred from the teacher model. As a result, the proposed model achieves superior performance and delivers more consistent outcomes than the baseline models.

To further verify the stability and effectiveness of the proposed model, we visualized the learning curves during training, as shown in Figure 2 and Figure 3.

As depicted in Figure 2, the model achieves rapid accuracy improvement within the first 45 epochs and gradually converges. The training and testing accuracy curves remain closely aligned, indicating strong generalization and resistance to overfitting.

In Figure 3, the loss curves for both training and testing sets show a fast decline and converge near zero within approximately 40 epochs. The consistent trends between the two curves further confirm that the model optimization is stable and that the proposed dynamic knowledge distillation framework is effective.

We conducted statistical significance tests between our proposed method and all baseline methods across both datasets to ensure the robustness and reliability of our results. Specifically, we applied paired *t*-tests to compare the performance metrics, and the results confirm that the improvements achieved by our method are statistically significant (*p* < 0.05). These findings further validate the effectiveness of our approach and support our claim of superior performance.

### 4.5. Ablation Study

To further evaluate the effectiveness of the multimodal fusion module and the dynamic feedback regulation mechanism in the unimodal student model, this study conducted an ablation study. The purpose of the ablation study was to assess the contribution of each proposed component by incrementally adding or removing features. The following variations were considered:**Variation 1:** Excluding multimodal fusion features and the feedback regulation mechanism. In this case, the Hinton-KD method is used as a baseline for knowledge distillation.**Variation 2:** Incorporating multimodal fusion features for distillation but without applying the feedback regulation mechanism.**Variation 3:** Including both multimodal fusion features and the feedback regulation mechanism.

This study builds upon the original knowledge distillation framework [22] and incorporates multimodal fusion features and feedback regulation mechanisms to assess their effectiveness. Table 3 demonstrates that integrating multimodal fusion features significantly enhances the performance of the unimodal student model. This improvement is attributed to the learning of multimodal distribution features, which strengthen the student model’s generalization ability.

Furthermore, the introduction of the feedback regulation mechanism enhances the model’s performance by dynamically adjusting the teacher model based on the evolving characteristics of the student model, thereby promoting more efficient learning.

Additionally, this study explored the impact of data-splitting strategies on model performance. It was found that some models and methods achieved artificially high results on the DEAP dataset due to improper data preprocessing, which introduced severe biases. Specifically, when segments from the same trial appeared in both the training and test sets, the model could memorize trial-specific patterns, leading to inflated performance metrics.

To address this, this study employed strict data-splitting protocols to ensure no data leakage between the training and test sets. Under improper splitting, the multimodal teacher model achieved accuracies of 95.23% and 95.92% for valence and arousal, respectively. However, these results are not reflective of real-world scenarios. Therefore, strict preprocessing and evaluation methods were implemented to ensure realistic and reliable performance metrics.

## 5. Conclusions

This paper proposes a novel emotion recognition method that integrates a multimodal fusion module and a dynamic feedback regulation mechanism based on interactive features. By leveraging multimodal data and the transfer of interactive and heterogeneous features, the proposed approach effectively improves the performance of unimodal student models. The method addresses the challenges of emotion recognition using physiological signals such as EEG and EOG while enhancing the model’s performance through dynamic feedback refinement. Experimental results demonstrate that our method achieves superior performance and reduces the reliance on EEG signals, thereby expanding the applicability of emotion recognition models to broader scenarios. By focusing on EOG signals, this method widens the application scope of emotion recognition tasks and facilitates further advancements in the field. In summary, this method has significant potential for future development and application in emotion recognition. However, there is still room for improvement in several areas. This study considers only EEG and EOG signals as part of the multimodal input. In the future, incorporating additional modalities could lead to more comprehensive and robust multimodal emotion recognition models. Moreover, the proposed framework is based on a dual-stream transformer encoder architecture, which demands substantial computational resources. Exploring lightweight models for efficient real-time computation and deployment on edge devices remains a critical challenge. Despite these limitations, the proposed method represents a major improvement over current state-of-the-art models. Future research could focus on developing interpretable neural networks for physiological signal encoding and decoding, which remains a crucial direction for physiological signal classification. Future work will incorporate additional modalities such as GSR and facial expressions to enrich the multimodal feature space. We will consider structural pruning, compression after distillation, and deployment based on lightweight models in the future. And combining Bayesian deep learning can also introduce more interpretability into deep models in the future, which will contribute to the development of emotion recognition based on physiological signals.

## Figures and Tables

**Figure 1 brainsci-15-00707-f001:**
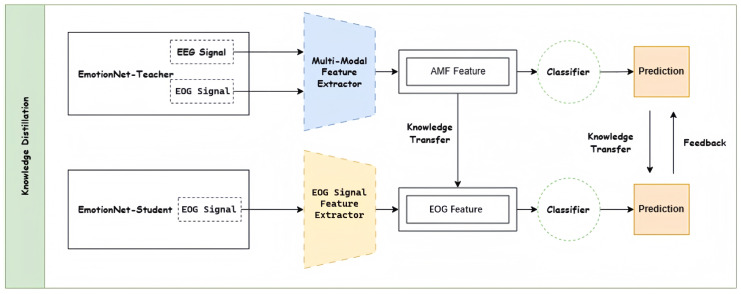
The overall framework of multimodal knowledge distillation.

**Figure 2 brainsci-15-00707-f002:**
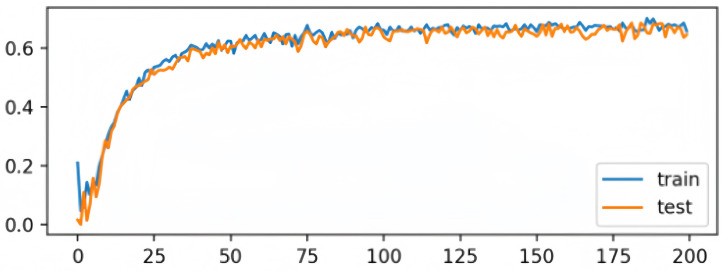
Accuracy curve during training. The blue line represents training accuracy, and the orange line represents testing accuracy. The model shows rapid improvement during the first 45 epochs and gradually stabilizes, with training and testing accuracy closely aligned—indicating good generalization performance.

**Figure 3 brainsci-15-00707-f003:**
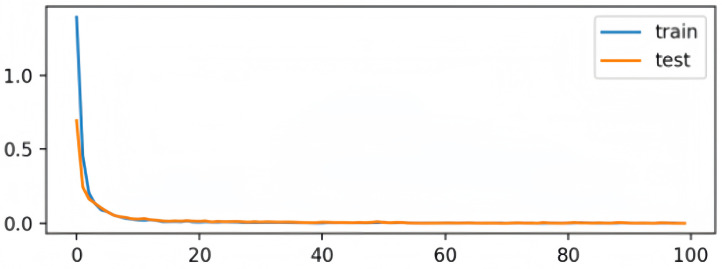
Loss curve during training. Both training and testing loss values decrease rapidly and converge smoothly, suggesting a stable optimization process and no signs of overfitting.

**Table 1 brainsci-15-00707-t001:** Comparison of proposed unimodal student model and baseline models on DEAP.

	Aro		Val	
	Accuracy	F1-Score	Accuracy	F1-Score
DeepConvNet [38]	54.70	51.95	66.45	61.15
CNN+RNN [30]	54.17	40.37	67.97	64.17
CGAN [39]	54.43	48.82	55.17	35.66
EmotionNet [38]	56.06	55.50	69.18	68.33
EmotionKD [40]	56.56	56.18	69.38	68.63
HKD-MER [41]	58.75	58.38	69.73	69.03
Proposed Method	60.41	61.01	70.38	70.16

**Table 2 brainsci-15-00707-t002:** Comparison of proposed unimodal student model and baseline models on BJTU-Emotion.

	Aro		Val	
	Accuracy	F1-Score	Accuracy	F1-Score
DeepConvNet [38]	56.56	42.55	46.32	48.69
CNN+RNN [30]	48.90	44.77	57.25	49.69
CGAN [39,42]	55.43	45.82	58.17	39.66
EmotionNet [38]	57.86	51.49	58.56	58.81
EmotionKD [40]	58.66	55.68	60.38	59.66
HKD-MER [41]	58.85	58.86	60.78	60.06
Proposed Method	60.41	59.81	61.31	60.61

**Table 3 brainsci-15-00707-t003:** Ablation study results on the DEAP dataset for different experimental configurations.

	Aro		Val	
	Accuracy	F1-Score	Accuracy	F1-Score
Variation 1	54.68	53.31	68.91	67.39
Variation 2	55.89	53.60	69.70	68.41
Variation 3	56.15	54.49	69.21	69.41

## Data Availability

The DEAP dataset used in this study is publicly available in the DEAP Dataset repository, https://www.eecs.qmul.ac.uk/mmv/datasets/deap/, accessed on 29 May 2025. The BJTU-Emotion dataset used in this study is available on request from the corresponding author due to privacy concerns regarding participant information.

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
