# Peer review of "Multimodal Knowledge Distillation for Emotion Recognition"

_brainsci, 2025, doi:10.3390/brainsci15070707_

Round 1
Reviewer 1 Report (Previous Reviewer 2)
Comments and Suggestions for Authors
The authors have addressed my comments.
Author Response
Please see the attachment.

Reviewer 2 Report (Previous Reviewer 4)
Comments and Suggestions for Authors
The author has already solved all of my problems. And the authors also conducted some additional experiments. I suggest accepting this paper.
Author Response
Please see the attachment.

Reviewer 3 Report (Previous Reviewer 3)
Comments and Suggestions for Authors
Despite the revisions, the manuscript still appears to conflate Electrooculography (EOG) and Galvanic Skin Response (GSR) in the context of emotion recognition. These techniques are conceptually and physiologically distinct: EOG primarily measures eye movements and blink activity, while GSR captures changes in skin conductance linked to autonomic arousal. Although both can be used as features in affective computing, their underlying mechanisms and typical applications differ significantly. Throughout the manuscript, there are instances where these techniques are treated as interchangeable or where their respective roles in emotion detection are not clearly delineated. I strongly encourage the authors to revise these sections to reflect a more accurate and differentiated understanding of EOG and GSR, both in the theoretical framing and in the interpretation of results.
Round 2
Reviewer 3 Report (Previous Reviewer 3)
Comments and Suggestions for Authors
The manuscript is clear in the press form, thus letting the reader fully appreciate the proposed approach.
One minority:
Line 217: attention mechanism Attention should be Attention mechanism
This manuscript is a resubmission of an earlier submission. The following is a list of the peer review reports and author responses from that submission.
Round 1
Reviewer 1 Report
Comments and Suggestions for Authors
This paper proposes a multimodal knowledge distillation framework for emotion recognition using EEG and EOG signals. Here are my comments on the given manuscript:
- The abstract section is incomplete and does not concisely summarize the experimental setup or concrete results.
-This paper lacks the novelty. The concept of distilling multimodal knowledge into a unimodal model is not new and has been previously studied. The originality of the proposed dynamic feedback mechanism is underexplored, and no comparisons are made with recent distillation baselines to substantiate its added value.
Muti-Modal Emotion Recognition via Hierarchical Knowledge Distillation
EmotionKD: A Cross-Modal Knowledge Distillation Framework for Emotion Recognition Based on Physiological Signals
-Missing Baseline Comparisons: The selected baselines (DeepConvNet, EmotionNet, CNN+RNN) are outdated. No comparison is made with recent Transformer-based models or knowledge distillation methods in multimodal emotion recognition.
- Please provide more experimental results for your proposed model, including additional figures for loss curves, learning curves and confusion matrices.
-Limitations of using only EEG and EOG (why not include GSR, facial cues?).
- The manuscript contains numerous grammatical and typographical errors.
- No consideration is given to model interpretability, computational complexity, or feasibility of real-time deployment—topics that are increasingly important in emotion recognition using physiological signals.
Reviewer 2 Report
Comments and Suggestions for Authors
The manuscript proposes an emotion recognition framework using multimodal knowledge distillation (EEG + EOG) to enhance the performance of unimodal models. While the topic is timely and relevant, several issues hinder the manuscript’s clarity and contribution. The core ideas have potential, but the writing quality, depth of literature integration, and result presentation need substantial improvement before the paper is suitable for publication.
- The Related Work section references several older or generic studies. Please expand the literature review and incorporate at least 5–8 recent studies (past 2 years) specifically targeting emotion recognition using physiological signals.
- The Results section lacks visual representation.
- Numerical improvements are reported without statistical significance or confidence intervals.
- Limited discussion on why proposed methods outperform the baselines or the effect of each module in practical terms.
- Please add more insight into how dynamic feedback actually contributes beyond numbers. I hope you do this for all your claims in the ablation study.
- Frequent grammar issues, unclear phrasing, and inconsistent technical terms.
- Equations are presented with repetitive descriptions and without strong intuitive interpretation.
- Please replace redundant text with deeper insights or expanded experiments.
Reviewer 3 Report
Comments and Suggestions for Authors
Unfortunately the whole paper is based on a confusion and inter-changable use of EOG sometimes as galvanic skin respone (GSR), sometimes as electrooculography (EOG).
Starting from the abstract, the Introduction and the related work sections, up to the Experimental results and analysis and Conclusion sections, this confusion prevents from getting real access to the proposed method.
My suggestion is to clarify which kind of measures you are dealing with and to resubmit a revised version of the paper.
I can only suppose that you are taking into account galvanic skin respone (GSR). However, it should be clearly and undoubtly defined from nthe authors, since my assumption based on the fact that EOG measures eye movements by detecting the corneo-retinal standing potential that exists between the front and the back of the human eye. Thus, EOG provides measures for saccades, fixations, blinks, attention information, being however less directly related to emotional states, with the only exception of fatigue and evetual mental workload in executing a task.
Reviewer 4 Report
Comments and Suggestions for Authors This paper addresses the problem of emotion recognition driven by physiological signals and proposes a framework based on multimodal knowledge distillation. The framework integrates EEG and EOG signals for joint modeling and distills the multimodal interactive and heterogeneous information into a unimodal EOG model, thereby improving its applicability in real-world scenarios. Experiments conducted on the DEAP and BJTU-Emotion datasets demonstrate that the proposed method outperforms existing approaches. 1. The specific architecture details of the “heterogeneous feature extractor” and “interaction extractor” modules (e.g., the dimensions of fully connected layers, activation functions, etc.) are not provided. 2. Non-standard spellings such as “stu-model” and “teather-model” are used throughout the paper. These should be standardized to “student model” and “teacher model” to align with common terminology. 3. The authors should compare their work with some recent studies on knowledge distillation and neural signal decoding. 4. The related work section does not adequately reflect recent advances in the fields of knowledge distillation and affective computing.